# Meta Adversarial Training against Universal Patches

**Jan Hendrik Metzen** [1]   **Nicole Finnie** [1]   **Robin Hutmacher** [1]

## Abstract

Recently demonstrated physical-world adversarial attacks have exposed vulnerabilities in perception systems that pose severe risks for safety-critical applications such as autonomous driving. These attacks place adversarial artifacts in the physical world that indirectly cause the addition of a universal patch to inputs of a model that can fool it in a variety of contexts. Adversarial training is the most effective defense against image-dependent adversarial attacks. However, tailoring adversarial training to universal patches is computationally expensive since the optimal universal patch depends on the model weights which change during training. We propose meta adversarial training (MAT), a novel combination of adversarial training with meta-learning, which overcomes this challenge by meta-learning universal patches along with model training. MAT requires little extra computation while continuously adapting a large set of patches to the current model. MAT considerably increases robustness against universal patch attacks on image classification and traffic-light detection.

## 1. Introduction

Deep learning is currently the most promising method for open-world perception tasks such as in automated driving and robotics. However, the use in safety-critical domains is questionable, since a lack of robustness of deep learning-based perception has been demonstrated (Szegedy et al., 2014; Goodfellow et al., 2015; Metzen et al., 2017; Hendrycks & Dietterich, 2019). *Physical-world* adversarial attacks (Kurakin et al., 2017; Athalye et al., 2018; Braunegg et al., 2020) are one of most problematic failures in robustness of deep learning. In this work, we focus on one subset of these physical-world attacks, where an attacker places a printed pattern in a scene that does not overlap with the tar-

[1]Bosch Center for Artificial Intelligence. Correspondence to: Jan Hendrik Metzen <janhendrik.metzen@de.bosch.com>.

*Accepted by the ICML 2021 workshop on A Blessing in Disguise: The Prospects and Perils of Adversarial Machine Learning.* Copyright 2021 by the author(s).

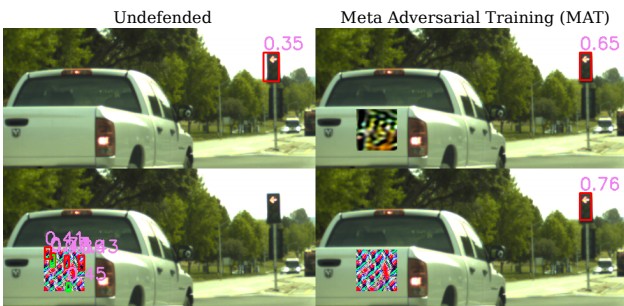

Undefended          Meta Adversarial Training (MAT)

*Figure 1.* Illustration of a digital universal patch attack against an undefended model (left) and a model defended with meta adversarial training (MAT, right) on Bosch Small Traffic Lights (Behrendt & Novak, 2017). A patch can lead the undefended model to detect non-existent traffic lights and miss real ones that would be detected without the patch (bottom left). In contrast, the same patch is ineffective against MAT (bottom right). Moreover, a patch optimized for MAT (top right), which bears a resemblance to traffic lights, does not cause the model to remove correct detections.

get object (Lee & Kolter, 2019; Huang et al., 2019).We study approaches for increasing robustness against such attacks in the strictly stronger threat model of universal adversarial *digital-domain* patch attacks (Brown et al., 2017).

The most promising method for increasing robustness against general adversarial attacks is adversarial training (Goodfellow et al., 2015; Madry et al., 2018), which simulates an adversarial attack for every mini-batch and trains the model to become robust against such an attack. Adversarial training against universal patches (and universal perturbations in general) is complicated by the fact that generating such universal patches is computationally much more expensive than generating image-dependent perturbations. Existing approaches for tailoring adversarial training to universal perturbations either refrain from simulating attacks in every mini-batch (Moosavi-Dezfooli et al., 2017; Hayes & Danezis, 2018; Perolat et al., 2018), which bears the risk that the model easily overfits these fixed or rarely updated universal perturbations, or alternatively use proxy attacks that are computationally cheaper such as "universal adversarial training" (UAT) (Shafahi et al., 2018) and "shared adversarial training" (SAT) (Mummadi et al., 2019). The latter face the challenge of balancing the implicit trade-off between simulating universal perturbations accurately and keeping computation cost of proxy attacks small.

We propose meta adversarial training (MAT) [1], which falls into the category of proxy attacks. MAT combines adversarial training with meta-learning. We summarize the key novel contributions of MAT and refer to Sec. 3 for details:

- MAT shares information about optimal patches over consecutive steps of model training in the form of meta-patches. These meta-patches allow generating strong approximations of universal patches with few iterations. In contrast to UAT (Shafahi et al., 2018), MAT uses meta-learning for sharing of information rather than joint training, which empirically generates stronger patches and a more robust model.
- MAT meta-learns a large set of meta-patches concurrently. While a model easily overfits a single meta-patch, even if it changes as in UAT, overfitting is much less likely for larger sets of meta-patches as in MAT.
- MAT encourages diversity of meta-patches by assigning a random but fixed target class and step-size to each meta-patch. This avoids that many meta-patches exploit the same model vulnerability.

We refer to Figure 1 for an illustration of MAT for universal patch attacks against traffic light detection. We note that while we focus on universal patches, MAT can be easily applied to universal $\ell_p$-norm pertubations (Sec. D.1.2).

## 2. Background and Related Work

Let $\mathcal{D}$ be a distribution over $d$-dimensional datapoints $x \in [0,1]^d$ and corresponding labels $y$, $\theta$ model parameters to be optimized, and $\mathcal{L}$ a loss function. Moreover, let $\mathcal{S}$ be the set of allowed perturbations and $\mathcal{F}$ be a function that applies a perturbation $\xi \in \mathcal{S}$ to a datapoint, potentially dependent on the label and some randomness $r \sim \mathcal{R}$. In the case of adversarial patches, $\mathcal{F}$ corresponds to overwriting a randomly chosen input region by a universal patch $\xi$ of a given size.

Following the notation introduced by Mummadi et al. (2019), we define the universal adversarial risk as follows: $\rho_{uni}(\theta) = \max_{\xi \in \mathcal{S}} \mathbb{E}_{(x,y) \sim \mathcal{D}, r \sim \mathcal{R}} \mathcal{L}(\theta, \mathcal{F}(x, \xi, r), y)$, where we drop the explicit dependence of $\rho_{uni}$ on $\mathcal{S}$, $\mathcal{D}$, and $\mathcal{R}$. Generally, we are interested in finding model parameters that minimize the universal adversarial risk, denoted as $\theta^* = \arg\min_\theta \rho_{uni}(\theta)$. This corresponds to the standard min-max saddle point formulation of adversarial training introduced by Madry et al. (2018), where we incrementally update the model parameters $\theta$ by computing $\theta_{t+1}$ based on $\nabla_{\theta_t} \rho_{uni}(\theta_t)$ (or more precisely an approximation of $\rho_{uni}(\theta_t)$). However, in contrast to standard adversarial training, the inner maximization problem is optimized over an expected value with respect to the data distribution $\mathcal{D}$

[1] https://github.com/boschresearch/meta-adversarial-training

and potential randomness $\mathcal{R}$, making it more expensive to solve (even approximately). As the optimal $\xi_t$ of the inner maximization at step $t$ of the outer minimization depends on the model parameters $\theta_t$, this maximization of $\xi_t$ needs to be repeated in every step of the outer minimization, making the direct minimization of $\rho_{uni}(\theta)$ intractable.

Existing work has addressed this in different ways (see Sec. H for a more detailed review of related works). One approach (Moosavi-Dezfooli et al., 2017; Hayes & Danezis, 2018; Perolat et al., 2018) relaxes the explicit dependence of $\xi_t$ on $\theta_t$ and computes a set or distribution over $\xi$ for some parameter checkpoints of $\theta$, and then applies these perturbations to the model while updating its parameters $\theta$. This can easily result in overfitting the fixed set or distribution over $\xi$. Another approach proposed by Mummadi et al. (2019) replaces the distribution $\mathcal{D}$ in the inner maximization with the current batch of the outer minimization. The effectiveness of this procedure hinges on the ability to efficiently approximate this inner maximization with few gradient steps.

## 3. Meta Adversarial Training

We propose a combination of adversarial training with meta-learning for increasing robustness against universal patches.

**Meta-learning Universal Patches.** As discussed in Sec. 2, the main challenge of using adversarial training to increase model robustness against universal patches is efficiently approximating $\xi_t$ for the current $\theta_t$ in every step $t$ of the outer minimization. Similar to UAT (Shafahi et al., 2018), we exploit the property that one step of the outer minimization only applies a small change to $\theta$; thus, for consecutive steps $t$ and $t + 1$ of the outer minimization, the resulting inner maximization problems for finding $\xi_t$ and $\xi_{t+1}$ are closely related (Zheng et al., 2020). UAT exploits this property by initializing the inner maximization at $t + 1$ with the (approximate) solution for $\xi_t$ and performs a single gradient step on a single batch in the inner maximization at $t + 1$. A potential shortcoming of this method is that it uses only a single gradient-step and thus implements joint training of parameters and patch, which does not allow capturing higher-order derivatives of the loss function (Nichol et al., 2018) and may therefore learn suboptimal initial patches.

We address this shortcoming by meta-learning initial values for universal patches – by approaching the optimization problems $\{\xi_{t_i}\}_{i=1}^N$ with gradient-based meta-learning: in parallel to updating $\theta_t$ in the outer minimization, we meta-learn an initialization $\Xi_t$, which we refer to as the "meta-patch" at time step $t$ of the outer minimization. That allows for approximating the inner-optimization problem of $\xi_{t+1}$ with few gradient steps. More precisely, we use the REPTILE (Nichol et al., 2018) meta-learning al-

gorithm with the iterative fast gradient sign method (I-FGSM) (Kurakin et al., 2017) task learner. In the inner maximization, we employ $K$ iterations of I-FGSM with $\xi_t^{(0)} = \Xi_t$ and for $k \in \{1, \ldots K\}$ the update rule $\xi_t^{(k+1)} = \Pi_{\mathcal{S}} \left[ \xi_t^{(k)} + \alpha \operatorname{sgn}(\nabla_\xi \mathcal{L}(\theta_t, \mathcal{F}(x, \xi_t^{(k)}, r), y)) \right]$, where $\Pi_{\mathcal{S}}$ denotes projection on the set $\mathcal{S}$ and $\alpha$ the step size of I-FGSM. The key difference compared to standard I-FGSM and PGD (Madry et al., 2018) is that the initialization $\xi_t^{(0)}$ is neither constant nor randomly sampled but meta-learned. The resulting patch $\xi_t^{(K)}$ is used two-fold: first, it is used with the REPTILE meta-learner for updating $\Xi$ with the following update: $\Xi_{t+1} = (1 - \sigma)\Xi_t + \sigma \xi_t^{(K)}$, where $\sigma$ is the learning rate of REPTILE. Second, $\xi_t^{(K)}$ is used in the next step of the outer minimization as an approximation of the optimal $\xi_t$ for the sample $(x, y)$ with randomness $r$. Learning the universal perturbation in UAT can be seen as a special case of our procedure for $K = 1$ and $\sigma = 1$.

**Meta-learning Diverse Collections of Patches.** The procedure proposed above allows meta-learning of a single meta-patch $\Xi$. However, one such meta-patch can easily get trapped in a local optimum, from which gradient-based meta-learning cannot easily escape. To prevent this, we propose a meta-learning approach that learns an entire set $\mathcal{P}$ of $P$ meta-patches $\Xi_i$. For each sample, we select one of these meta-patches that will be used for initializing I-FGSM and later get updated by REPTILE. However, meta-learning a set of meta-patches in this way with the same optimizer and objective will not automatically result in a diverse set of meta-patches. We encourage diversity of the generated set of meta-patches by randomly assigning a target and performing a targeted I-FGSM attack. This avoids many patches converging to similar patterns that fool the model into predicting the same class. Moreover, we also assign a randomly chosen fixed step size $\alpha$ for I-FGSM to every meta-patch. Larger step sizes correspond to meta-patches that explore the space of allowed patches more globally while smaller step sizes result in more fine-grained attacks. We evaluate the effectiveness of these heuristics in Sec. 4.

**Meta Adversarial Training (MAT).** We summarize MAT in Algorithm 1 (see Section A.3 for additional considerations). The function $\operatorname{INIT}^P$ (Algorithm 2 in the appendix) initializes $\mathcal{P}$ consisting of $P$ meta-patches $\Xi_i$ and corresponding targets $y_i^{target}$ and step-sizes $\alpha_i$. We select the target as one of the classes in a round-robin fashion and the step size log-uniformly from $[0.0001, 0.1]$. We initialize the meta-patches by either sampling uniform randomly from $[0, 1]^d$ or by (sub-sampling) an actual data-point, which corresponds to an on-manifold initialization akin to CutMix (Yun et al., 2019). This data-initialization was concurrently proposed by Yang et al. (2020b), and Yang et al. (2020a) found that such texture patches can be adversarial even without further optimization. The function $\operatorname{SELECT}^F$ (Al-

---

**Algorithm 1** Meta Adversarial Training

1: **Input:** data $\mathcal{D}$, initial parameters $\theta$, application fct. $\mathcal{F}$, loss-fct. $\mathcal{L}$, REPTILE learn-rate $\sigma$
2: # Initialize $P$ tuples of meta-patch, target, and step size
3: $\mathcal{P} \leftarrow \operatorname{INIT}^P(\mathcal{D}, \text{"data"})$
4: # $t_{max}$ steps of outer minimization (we drop subscript $t$)
5: **for** $t$ in $\{0, \ldots, t_{max} - 1\}$ **do**
6:    # Sample datapoint
7:    $(x, y) \sim \mathcal{D}$
8:    # Select meta-pert. $\Xi$ and corresponding target $y^{target}$, step-size $\alpha$, and randomness $r$ from $\mathcal{P}$
9:    $\Xi, y^{target}, \alpha, r \leftarrow \operatorname{SELECT}^F(\mathcal{P}, x, y, \theta_t, \mathcal{F}, \mathcal{L}, \mathcal{R})$
10:   # Inner maximization initialized with meta-pert. $\Xi$
11:   $\xi \leftarrow \operatorname{I-FGSM}^K(\Xi, x, y^{target}, \theta_t, \mathcal{F}, \alpha, r)$
12:   # Outer minimization step with optimizer OPT, e.g., SGD
13:   $\theta \leftarrow \operatorname{OPT}(\mathcal{L}, \theta, \mathcal{F}(x, \xi, r), y)$
14:   # Meta-learning update of meta-patch $\Xi$ (REPTILE)
15:   $\Xi \leftarrow (1 - \sigma)\Xi + \sigma \xi$
16:   # Replace updated meta-patch in $\mathcal{P}$
17:   $\mathcal{P} \leftarrow \operatorname{UPDATE}(\mathcal{P}, \Xi)$
18: **end for**

---

gorithm 3 in the appendix) uniform randomly samples $F$ trials of $(\Xi, y^{target}, \alpha) \sim \mathcal{P}$ with randomness $r \sim \mathcal{R}$ and selects the one with maximizal loss $\mathcal{L}(\theta, \mathcal{F}(x, \Xi, r), y)$.

Line 11-15 present the core of MAT consisting of (i) inner maximization of a patch $\xi$ that was initialized from a meta-patch $\Xi$ with I-FGSM$^K$, (ii) a step of outer minimization of $\theta_t$ with an optimizer like SGD on a pair of perturbed input $\mathcal{F}(x, \xi, r)$ and corresponding label $y$, and (iii) the meta-learning update of the respective meta-patch $\Xi$ with REPTILE. We can easily extend Algorithm 1 to a batch size larger than one: the only required change is that REPTILE-based meta-learning deals with the situation where the same meta-patch is selected and optimized for several elements in a batch. In this case, the meta-learning update becomes $\Xi \leftarrow (1 - \sigma)\Xi + \sigma \frac{1}{N} \sum_{i=1}^N \xi_i$ for $N$ patches $\xi_i$ initialized from the same meta-patch $\Xi$. We note that MAT has similar computational cost as adversarial training (Sec. A.2).

## 4. Experiments

We conduct experiments on patch robustness of models for image classification and for object detection. We refer to Sec. B for details on attacks used for robustness evaluation and to Sec. C for experimental details.

### 4.1. Image Classification on Tiny ImageNet (Tin)

We evaluate robustness against universal patches of size 24x24 pixel that cover approx. 14% of the 64x64 images. Patches are randomly translated from the center of the image by at most 26 pixels. Results are summarized in Table 1 (more details can be found in Sec. D.1). We observe that a model trained with standard empirical risk minimization offers no robustness against any of the evaluated attacks. When comparing the baseline adversarial defenses (Madry

*Table 1.* Accuracy (mean over 5 runs) on Tiny ImageNet on clean data (CL) and against universal patch attacks with random init (RI), init with a cropped image patch (DI), low-frequency filter (LF), transfer attacks (Tr), and worst across all four attacks (Min). Shown are adversarial training (AT) (Madry et al., 2018), shared adversarial training (SAT) (Mummadi et al., 2019), universal adversarial training (UAT) (Shafahi et al., 2018), and MAT ablations.

| Setting | CL | RI | DI | LF | Tr | Min |
|---|---|---|---|---|---|---|
| Standard | 0.55 | 0.03 | 0.03 | 0.07 | 0.03 | 0.02 |
| AT | 0.57 | 0.06 | 0.07 | 0.12 | 0.17 | 0.06 |
| SAT | 0.58 | 0.07 | 0.07 | 0.12 | 0.33 | 0.06 |
| UAT | 0.49 | 0.41 | 0.09 | 0.15 | 0.12 | 0.09 |
| MAT (FULL) | **0.59** | 0.56 | 0.55 | 0.53 | **0.55** | **0.53** |
| (RAND. INIT) | 0.58 | 0.56 | 0.32 | 0.26 | 0.52 | 0.23 |
| (UNTARGET) | 0.58 | 0.56 | **0.56** | **0.56** | 0.50 | 0.50 |
| ($F = 1$) | 0.58 | 0.53 | 0.53 | 0.50 | 0.52 | 0.48 |
| ($K = 1$) | 0.58 | 0.55 | 0.53 | 0.53 | 0.46 | 0.45 |
| ($\sigma = 1.0$) | 0.58 | **0.57** | 0.55 | 0.55 | 0.50 | 0.50 |

*Table 2.* Mean recall (mR) and mean average precision (mAP) of different methods on clean data and against universal patch attacks (Adv) on Bosch Small Traffic Lights (Behrendt & Novak, 2017).

| | MR | Adv MR | MAP | Adv MAP |
|---|---|---|---|---|
| Standard | **0.47** | 0.37 | **0.41** | 0.09 |
| UAT | **0.47** | **0.45** | **0.41** | 0.26 |
| MAT (DEFAULT) | 0.45 | 0.43 | **0.41** | 0.35 |
| (+DATA INIT) | 0.45 | 0.42 | **0.41** | **0.38** |
| ($K = 1$) | 0.40 | 0.43 | **0.41** | 0.27 |
| ($K = 1, P = 1$) | 0.46 | 0.43 | **0.41** | 0.17 |

et al., 2018; Mummadi et al., 2019; Shafahi et al., 2018), none of them exhibit more than trivial robustness. We note that while UAT provides relatively high robustness against attacks with random initialization, this robustness does not carry over to other attack variants.

In contrast, MAT (full) with standard parameters (INIT with data initialization and $P = 1000$ meta-patches, targeted attacks, $F = 5$ in SELECT, $K = 5$ iterations in I-FGSM, REPTILE learning rate $\sigma = 0.25$) shows high robustness against all attack variants. When ablating MAT, choosing a random initialization in INIT is most problematic – it results in similar but less severe overfitting to randomly initialized attacks as UAT. Also, ablating towards joint training ($\sigma = 1.0$ and $K = 1$) deteriorates performance relative to meta-learning in MAT (full). In addition, enforcing diversity in meta-patches via targeted attacks in MAT (full) is responsible for a small increase in robustness compared to untargeted learning of meta-patches. Finally, taking the worst over $F = 5$ samples in SELECT outperforms random sampling ($F = 1$).

Importantly, MAT offers increased robustness without affecting clean performance. In contrast, MAT acts as an effective regularizer and reduces overfitting compared to standard training and achieves the strongest clean performance among all methods, surpassing standard training by 4 percentage points. Even against the strongest patches, MAT only loses 2 percentage points accuracy relative to standard training on clean data, despite the relatively large patch size. We provide illustrations of patches in Sec. E.

### 4.2. Object Detection on Bosch Small Traffic Lights

We evaluate robustness of a traffic light detector based on YoloV3 (Redmon & Farhadi, 2018) trained on the Bosch

Small Traffic Lights Dataset (Behrendt & Novak, 2017). We add 64x64 patches, covering 0.45% of the 1280x704 images, and add random translations from the center by up to $(512, 282)$ pixels. We evaluate performance using mean Average Precision (mAP) and mean Recall (mR) over classes for a fixed confidence threshold. While mAP captures both non-existent detections caused by the patch (false positives) and correct detections missed by the model (false negatives), mean recall focuses only on the latter, so called "blindness" attacks (Saha et al., 2019). For MAT (default), we use random initialization in INIT and $P = 10$, $F = 1$, $K = 3$, and REPTILE learning rate $\sigma = 0.25$. We refer to Sec. C.4 for more details on the experimental setting.

Table 2 summarizes the results (more details can be found in Sec. D.2). Standard training faces a drop in mR when a universal patch is added and is thus likely susceptible to physical-world blindness attacks. In contrast, UAT and all variants of MAT are very robust against the tested blindness attacks even in the digital domain. In terms of the mAP, UAT faces a considerable drop for patch attacks. A similar but weaker effect can also be observed for MAT with the default configuration. Both methods therefore detect non-existent traffic lights on the patch or in its vicinity. Interestingly, false positive detections often resemble traffic lights (see Figure 1 and Sec. F). MAT with data initialization in INIT, however, is very robust in terms of prevention of additional false positives as indicated by the high mAP. When ablating MAT, its mAP deteriorates as the configuration approaches UAT ($K = 1$ and $P = 1$). We conclude that all aspects of MAT are essential for achieving maximal robustness.

## 5. Conclusion

We propose meta adversarial training (MAT), a novel combination of adversarial training with meta-learning that allows the increase of model robustness against universal patches with little computational overhead. Moreover, we show that prior work, which was assumed to be robust, can be fooled by stronger attacks. In contrast, MAT remains robust against all evaluated attacks. Our results indicate that physical-world attacks will become considerably more difficult against models trained with MAT.

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

# META ADVERSARIAL TRAINING AGAINST UNIVERSAL PATCHES
## SUPPLEMENTARY MATERIAL

## A. Details on Meta Adversarial Training

### A.1. Summary of Advantages over Related Work

We briefly summarize the main advantages of MAT compared to prior work: as opposed to UAT (Shafahi et al., 2018), MAT meta-learns a diverse set of meta-patches with I-FGSM$^K$ concurrently to model training rather than jointly training model parameters and a single perturbation with FGSM. Compared to SAT (Mummadi et al., 2019), MAT does not treat every inner maximization problem independently but meta-learns strong initializers, allowing MAT to find stronger patches with no more computational cost than standard adversarial training (see Section A.2). In contrast to the work of Moosavi-Dezfooli et al. (2017); Hayes & Danezis (2018); Perolat et al. (2018), MAT computes novel patches in every iteration of model training (outer minimization). We would also like to note that MAT meta-learns patches but not model weights and thus results in a standard trained model that does not require test-time adaptation.

### A.2. Computational Cost of Meta Adversarial Training

Computational cost of MAT is dominated by the number of forward passes $n_{fp}$ and backward passes $n_{bp}$ through the network for a single iteration of model training (the outer loop). Adversarial training (AT) with $K$-step PGD incurs $(K + 1) * (n_{fp} + n_{bp})$ cost for one iteration: $K * (n_{fp} + n_{bp})$ for generating the patch and $1 * (n_{fp} + n_{bp})$ for one step of model training. Similarly, the cost of MAT (with REPTILE as in our experiments) for inner maximization and one step of outer minimization is $(K + 1) * (n_{fp} + n_{bp})$. Additionally, selecting a meta-patch from $F$ samples in Algorithm 3 incurs a cost of $F * n_{fp}$ for $F > 1$. The additional cost is 0 for $F = 1$ because the meta-patch is sampled randomly in this case and its loss need not be computed. The cost of REPTILE itself is negligible, because it is a simple convex combination. Therefore, the cost of MAT ($F = 1$) and that of AT are comparable and MAT ($F = 1$) clearly outperforms AT in Table 1. Also for a small $F$ such as $F = 5$ for MAT (full) in Table 1, MAT+REPTILE does not incur considerably higher cost than AT for the same $K$. The key point is that due to better initialization from the set of meta-patches, a small $K$ can be chosen for MAT, whereas $K$ would need to be very large for PGD in order to create equally strong attacks. Training MAT for Tiny ImageNet takes approximately 2 days on a single Tesla V100 SXM2 GPU, with the overhead compared to AT being one hour. For comparison, standard non-adversarial training takes approximately 11 hours on the same GPU.

### A.3. Estimating the Expected Loss and Variance Reduction

As outlined in Section 3, MAT is based on estimating the expected loss $\mathbb{E}_{(x,y)\sim\mathcal{D},r\sim\mathcal{R}}\mathcal{L}(\cdot)$. This estimate is required in every of the $K$ steps of I-FGSM as well as in the outer minimization step of updating $\theta_t$. We base this estimate per $\xi$ on a single sample $(x, y) \sim \mathcal{D}, r \sim \mathcal{R}$ for reasons of computational efficiency. Since this corresponds to a high variance estimate, we use the same sample in all $K$ steps of I-FGSM at time $t$ as well as in the outer minimization step of updating $\theta_t$. This provides us benefits of reduced variance and more efficient computation, however, at the cost of a biased estimate of $\rho_{uni}(\theta_t)$ – I-FGSM will converge to an $\xi_t^{(K)}$ that is overfit to $(x, y)$ and $r$. Compared to a patch optimized over the entire distributions $\mathcal{D}$ and $\mathcal{R}$, $\xi_t^{(K)}$ will incur a higher loss on the sample. Nevertheless, since we typically choose the number of I-FGSM steps $K \leq 10$, we expect only weak overfitting and the gains from reduced variance more than compensates for the increased bias in our experience.

## B. Reliable Robustness Evaluation

We outline strong attacks for reliably evaluating the robustness of trained models against universal perturbations and patches. Importantly, we do not use the meta-patches $\Xi_i$ as this might result in a biased robustness evaluation. Instead, we extend PGD (Madry et al., 2018) in a similar way as Mummadi et al. (2019) by rewriting $\rho_{uni}$ from Equation (2) to $\rho_{uni}(\theta) = \max_{\xi\in\mathcal{S}} \rho(\theta, \xi)$ with

$$\rho(\theta, \xi) = \underset{(x,y)\sim\mathcal{D},r\sim\mathcal{R}}{\mathbb{E}} \mathcal{L}(\theta, \mathcal{F}(x, \xi, r), y),$$

and then use the estimate

$$\hat{\rho}(\theta, \xi) = \frac{1}{N} \sum_{i=1}^{N} \mathcal{L}(\theta, \mathcal{F}(x_i, \xi, r_i), y_i)$$

based on samples $(x_i, y_i) \sim \mathcal{D}$ and $r_i \sim \mathcal{R}$. We define stochastic projected gradient descent (S-PGD) as $\xi^{(0)} \sim \mathcal{S}$ and $\xi^{(k)} = \Pi_{\mathcal{S}} \left[ \xi^{(k-1)} + \alpha \operatorname{sgn}(\nabla_\xi \hat{\rho}(\theta, \xi^{(k-1)})) \right]$. Note that S-PGD uses different $x_i, y_i, r_i$ in every step when estimating $\hat{\rho}$.

In general, S-PGD will converge to local optima; namely, $\xi^{(K)}$ obtained after K steps of S-PGD will not necessarily be the global maximizer of $\hat{\rho}(\theta, \xi)$. To account for this, we propose three extensions of S-PGD: Firstly, since the initialization of $\xi_0$ will generally affect the quality of $\xi^{(K)}$, we propose an alternative initialization akin to CutMix (Yun et al., 2019) where we initialize $\xi^{(0)}$ based on a datapoint $x \sim \mathcal{D}$. For universal patch attacks, we downsample or crop $x$ to the patch size, whereas for universal perturbation/patch attacks, we scale its intensity range such that $x \in \mathcal{S}$. This initialization becomes even more effective if we sample many $x \sim \mathcal{D}$ and select the one for initializing $\xi^{(0)}$ which would maximize $\hat{\rho}(\theta, \xi^{(0)})$. We denote this initialization as *data initialization*.

Secondly, we take inspiration from recently proposed *low-frequency attacks* (Guo et al., 2019; Sharma et al., 2019): we modify the process of adding a perturbation/patch $\xi$ to an input to $\mathcal{F}(x_i, LP(\xi, u), r_i)$, where $LP(\xi, u)$ denotes a low-pass filter with cutoff-frequency $u$. To achieve this, we follow Jo & Bengio (2017) and create a centered radial mask with radius $u$. The patch is transformed into frequency space and multiplied by the radial mask. The result is transformed back to image space and thus yields the patch to be applied to the image. While this makes the attack weaker in principle since only low-frequency perturbations/patches are possible, we observe that in practice, it can lead to a more well-behaved optimization problem and result in S-PGD converging to stronger perturbations/patches.

Thirdly, we perform a *transfer attack*, in which we run an attack after every epoch of model training. We initialize $\xi^{(0)}$ with one of the $\xi^{(K)}$ found in previous epochs, namely the one that would maximize $\hat{\rho}(\theta, \xi^{(0)})$. After every 5 epochs, we run an additional S-PGD attack from randomly initialized $\xi^{(0)}$. This transfer attack helps identify cases where universal perturbations/patches found in early epochs remain effective against the model but in later epochs are no longer found when running S-PGD attacks from random or data initialization.

# C. Implementation Details

## C.1. Subprocedure INIT$^P$

We present details on subprocedure INIT$^P$ in Algorithm 2. Two relevant parameters of INIT$^P$ are described below.

**Initialization of Meta-Patches** As described in Section 3, meta adversarial training (MAT) meta-learns a set $\mathcal{P}$ of $P$ meta-patches $\Xi^{(i)}$, where $i$ indexes $P$. Similar to the attack initialization described in Section B, these meta-patches can be initialized in INIT$^P$ in two ways as follows:

- *Random initialization*: sampling randomly from a uniform distribution over the space of allowed perturbations/patches

---

**Algorithm 2** INIT$^P$

1: **Input:** number of meta-perturb. $P$, data $\mathcal{D}$, initialization type "init"
2: $\mathcal{P} \leftarrow \{\}$
3: **for** $i$ **in** $\{1, \ldots, P\}$ **do**
4:     $y^{target} \leftarrow i \bmod C$ # round-robin target class, $C$ being the number of classes
5:     **if** init = "random" **then**
6:         $\Xi \sim \text{UNIFORM}([0, 1]^{d_{patch}})$
7:     **else if** init = "data" **then**
8:         $x \sim \text{UNIFORM}(\mathcal{D}|_{y=y^{target}})$ # Select datapoint labeled with target class uniformly
9:         $\Xi \leftarrow \text{RESIZE}(x, d_{patch})$ # Resize datapoint to appropriate dimensionality
10:     **end if**
11:     $\alpha \sim \text{LOGUNIFORM}(0.0001, 0.1)$
12:     $\mathcal{P} \leftarrow \mathcal{P} \cup \{(\Xi, y^{target}, \alpha)\}$
13: **end for**
14: Return $\mathcal{P}$

---

**Algorithm 3** SELECT$^F$

---

1: **Input:** number of trials $F$, set of meta-perturb. $\mathcal{P}$, input $x$, label $y$, parameters $\theta$, application fct. $\mathcal{F}$, loss-fct. $\mathcal{L}$, random generator $\mathcal{R}$
2: $\Xi_{opt}, y_{opt}^{target}, \alpha_{opt} \sim \text{UNIFORM}(\mathcal{P})$ # Sample uniform randomly from $\mathcal{P}$
3: $r_{opt} \sim \mathcal{R}$
4: **if** $F = 1$ **then**
5:    Return $(\Xi_{opt}, y_{opt}^{target}, \alpha_{opt}, r_{opt})$
6: **end if**
7: $l_{opt} \leftarrow \mathcal{L}(\theta, (\mathcal{F}(x, \Xi_{opt}, r_{opt}), y))$ # loss on perturbed data, according label for parameters $\theta$
8: **for** $i$ **in** $\{1, \ldots, F\}$ **do**
9:    # Find worst (in terms of loss) out of $F$ trials
10:    $\Xi, y^{target}, \alpha \sim \text{UNIFORM}(\mathcal{P})$
11:    $r \sim \mathcal{R}$
12:    $l \leftarrow \mathcal{L}(\theta, (\mathcal{F}(x, \Xi, r), y))$
13:    **if** $l > l_{opt}$ **then**
14:       $(\Xi_{opt}, y_{opt}^{target}, \alpha_{opt}, r_{opt}) \leftarrow (\Xi, y^{target}, \alpha, r)$
15:       $l_{opt} \leftarrow l$
16:    **end if**
17: **end for**
18: Return $(\Xi_{opt}, y_{opt}^{target}, \alpha_{opt}, r_{opt})$

---

$\mathcal{S}$.

- *Data initialization*: this initialization sub-samples actual data points from the training dataset and corresponds to an on-manifold initialization that follows the data distribution. To generate universal patches, we downsample or crop the data points. To create universal perturbations/patches, we scale the intensity of the data points to the range of $\mathcal{S}$.

**Number of Meta-Patches**   The number of meta-patches $P$ is chosen roughly proportional to the number of classes of the dataset regardless of classification or object detection tasks. We choose $P = 1000$ for Tiny ImageNet, which has 200 classes. For Bosch Small Traffic Lights Dataset, we choose $P = 10$ because the dataset only has 4 classes.

## C.2. Subprocedure SELECT$^F$

We present details on the sub-procedure SELECT$^F$ in Algorithm 3. We note that the special choice $F = 1$ corresponds to a uniform random sampling of a meta-patch (and corresponding target class, step-size, and randomness). For $F > 1$, SELECT$^F$ requires $F$ additional evaluations of the loss functions (and thus forward passes through the model) since the sample with the maximal loss is selected.

## C.3. Image Classification on Tiny ImageNet

To evaluate the robustness of a model trained with MAT against universal patch attacks, we compare its performance with other training approaches such as Standard, CutMix (Yun et al., 2019), PatchUniform, adversarial training (AT) (Madry et al., 2018), shared adversarial training (SAT) (Mummadi et al., 2019), and universal adversarial training (UAT) (Shafahi et al., 2018). Please note that this evaluation is an ablation study of MAT, namely, we configure MAT in a way that is similar to each training approach. Detailed configurations are shown for universal patches in Table 3 and for universal perturbations in Table 8.

We train every model for 75 epochs with SGD, an initial learning rate of 0.033, a cosine decay learning rate scheduler, momentum 0.9, and a batch size of 128. We use a ResNet (He et al., 2016), train it from scratch, and follow Xie & Yuille (2020) by replacing batch normalization with group normalization (Wu & He, 2019) and weight standardization (Qiao et al., 2019). We use $K = 5$ iterations of I-FGSM in AT (Madry et al., 2018), SAT (Mummadi et al., 2019), and MAT. For UAT (Shafahi et al., 2018), we use $K = 1$ following their recommendation. For SAT, we use sharedness 128. We note that all adversarial training baselines were trained against patch attacks.

For every setting, we perform 5 independent runs. We evaluate the robustness against 2500-step-S-PGD with a batch size of 64 and random initialization, data initialization (data samples resized to patch size), and low-frequency filter,

| SETTING | CUTMIX | PATCHUNIFORM | AT | SAT | UAT | MAT (FULL) |
|---|---|---|---|---|---|---|
| PATCH INITIALIZATION | DATA | RANDOM | RANDOM | RANDOM | RANDOM | DATA |
| WORST OVER $F$ SAMPLES | − | − | 1 | 1 | 1 | 5 |
| REPTILE LEARNING RATE $\sigma$ | − | − | 0 | 0 | 1 | 0.25 |
| NUMBER OF META-PATCHES $P$ | − | − | − | − | 1 | 1000 |
| I-FGSM STEP-SIZE $\alpha$ | − | − | [0.0001, 0.1] | 0.2 | 0.01 | [0.0001, 0.1] |
| I-FGSM ITERATIONS $K$ | − | − | 5 | 5 | 1 | 5 |
| SHAREDNESS | − | − | − | 128 | − | − |

*Table 3.* Configuration of training procedures against universal adversarial patch attacks for image classification. We denote irrelevant entries as '−'.

| Parameter | Values |
|---|---|
| Patch Initialization | random, data |
| Step Size $\alpha$ | 0.0001, 0.00033, 0.001, 0.0033, 0.01, 0.033, 0.1 |
| Momentum $\gamma$ | 0, 0.9, 0.99 |
| Cutoff Frequency | off, 12 |
| Number of iteration (S-PGD) | 2500 |
| Total Step Size Decay | 0.01 |
| Batch Size | 64 |

*Table 4.* Configuration grid of attacks against the classification tasks.

and the transfer attack (see Section B). For the S-PGD settings, we perform a grid search (see Table 4) over step sizes $\alpha \in \{0.0001, 0.00033, 0.001, 0.0033, 0.01, 0.033, 0.1\}$ and momentum $\gamma \in \{0, 0.9, 0.99\}$ independently for every trained model and report the minimal accuracy. Finally, we report the minimal accuracy across all attacks.

**Model Architecture**    For each setting, we train a ResNet V1 model (He et al., 2016) from scratch on the Tiny ImageNet dataset (Tin). This ResNet model contains 4 residual stacks, where each stack consists of 2 residual blocks. The stacks have 64, 128, 256, and 512 channels and spatial resolution of 64x64, 32x32, 16x16, 8x8, respectively. We employ ReLU as its activation function. Each convolutional layer has a stride of 1, kernel size of 3, group normalization with weight standardization, SAME padding, "he_normal" kernel initialization, and a weight decay of $1 \cdot 10^{-4}$ on the kernel weights.

**Model Training**    Each model is trained with 24x24 pixel patches applied to the 64x64 pixel input images, namely, a patch covers approximately 14% of the image. Patches are randomly translated from the center of the image by up to 26 pixels during training. We train each model with SGD for 75 epochs, an initial learning rate of 0.033, a cosine decay learning rate scheduler, momentum of 0.9, and a batch size of 128. For each setting, we perform 5 independent runs with 5 different seeds. Details regarding adversarial training procedures are shown in the Table 3. The most crucial parameter for AT, SAT, and UAT is the step size $\alpha$ of I-FGSM. For AT, we follow MAT and sample the step size per datapoint randomly from a log-uniform distribution over $[0.0001, 0.1]$. Since UAT and SAT only update a single patch per batch, this random sampling strategy is not feasible on a per-batch level. Instead, we use a fixed $\alpha$; more specifically, we use 0.2 for SAT such that I-FGSM can reach any value in $[0, 1]^d$ in K=5 iterations. Since UAT updates the perturbation/patch iteratively over the batches, a smaller value for $\alpha$ is feasible here and we employ $\alpha = 0.01$. We did not tune these choices for $\alpha$ extensively but note that MAT does not require any tuning of the step size $\alpha$.

**Model Evaluation**    As described in Section B, we propose strong attacks for reliably evaluating the robustness of trained models against universal patch attacks and universal perturbation attacks by optimizing the perturbations using S-PGD. We propose two initialization methods for S-PGD. Additionally, we utilize the low-frequency attack described in Section B. The S-PGD step size $\alpha$ is exponentially decayed with a total decay of 0.01. For evaluation of each model's robustness, we perform S-PGD attacks over the parameter grid given in Table 4. The attack results can be found in Subsection D.1.

### C.4. Object Detection on Bosch Small Traffic Lights Dataset

We describe the experimental details for training robust traffic light detectors against universal patch attacks.

| SETTING | UAT | MAT (DEFAULT) | MAT(+DATA) | MAT ($K$=1) | MAT($K$=1,$P$=1) |
|---|---|---|---|---|---|
| PATCH INITIALIZATION | RAND | RAND | DATA | RAND | RAND |
| WORST OVER $F$ SAMPLES | 1 | 1 | 1 | 1 | 1 |
| REPTILE LEARNING RATE $\sigma$ | 1.0 | 0.25 | 0.25 | 0.25 | 0.25 |
| NUMBER OF PATCHES $P$ | 1 | 10 | 10 | 10 | 1 |
| I-FGSM STEP-SIZE $\alpha$ | 0.01 | [0.0001, 0.1] | [0.0001, 0.1] | [0.0001, 0.1] | [0.0001, 0.1] |
| I-FGSM ITERATIONS $K$ | 1 | 3 | 3 | 1 | 1 |

*Table 5.* Configuration of MAT with different approaches against universal adversarial patch attacks for object detection.

| Parameter | Values |
|---|---|
| Patch Initialization | random (RI), data crop (DI) |
| Number of Steps (S-PGD) | 4000 |
| Batch Size | 4 |
| Step Size $\alpha$ | 0.1, 0.01, 0.001, 0.0001 |
| Total Step Size Decay | 0.01 |
| Momentum $\gamma$ | 0.9 |
| Cutoff Frequency $u$ | 25, 50, 100, 250 |
| Loss | standard, no objectness loss, ignoring false positives |

*Table 6.* Configuration grid of attacks against the detection tasks.

**Model Training**   For each training procedure, we train a Yolo V3 model (Redmon & Farhadi, 2018) from scratch on Bosch Small Traffic Lights Dataset (Behrendt & Novak, 2017). The model has three network outputs on each scale as implemented in the original paper. For each DarkNet conv layer, we replace batch normalization with group normalization and use weight standardization. To interpret the network outputs of Yolo V3, we set the confidence threshold to 0.3. This means only the predictions with an objectness score $> 0.3$ count as valid predictions. The non-maximum suppression threshold is set to 0.1, that means we prune the predictions when their bounding boxes overlap with IoU $> 0.1$.

Each model is trained with 64x64 pixel patches applied to the input images resized to 1280x704 – both width and height of the resized images are a multiple of 32 because a grid cell's size is 32x32; thus, a patch covers 0.45% of the image. Patches are randomly translated from the center of the image by up to (512, 282) pixels during training. We ensure that translated patches do not overlap with any ground-truth traffic-light annotation. We train the model with ADAM for 15 epochs, an initial learning rate of 0.0001, a cosine decay learning rate scheduler, and a batch size of 1. We compare the accuracy against universal patch attacks of UAT and MAT variants. The configuration details are shown in Table 5.

**Model Evaluation**   As described in Section 4.2, in order to evaluate the effectiveness of the universal patches as well as the robustness of the model, we apply two metrics to the evaluation procedure - mean Average Precision (mAP) and mean recall over classes with the IoU threshold of 0.1, which determines true positives between predicted bounding boxes and the ground truth. For generating universal patches, we use S-PGD with 4000 steps with a batch size of 4. To find the strongest patches, we perform a grid search over step sizes $\alpha \in \{0.1, 0.01, 0.001, 0.0001\}$, a fixed momentum $\gamma$ of 0.9, and a cutoff frequency $u \in \{25, 50, 100, 250\}$ of an optional low-pass filter. In addition, we also conduct a grid search over three different options for the loss that is maximized by the attacker - 1) the standard loss that is also used during training, 2) the standard loss subtracting the objectness loss, and 3) the standard loss ignoring all false positives. The last loss variant is also well suited for "blindness" attacks since it accentuates false negatives.

Similar to the previous initialization approaches in Section C.3, the perturbations/patches for these attacks are initialized in two different ways - randomly or from a cropped image of Bosch Small Traffic Lights Dataset. Each configuration is a unique combination of an initialization, a step size, a cutoff frequency, and a loss found through the grid search. The parameter grid is summarized in Table 6. More result details can be found in Section D.2.

# D. Result Details

## D.1. Image Classification on Tiny ImageNet

### D.1.1. ROBUSTNESS AGAINST UNIVERSAL PATCH ATTACKS

In Figure 2, we compare learning curves of MAT models against the transfer attack (see Section B) between different settings during training. The left plot shows that initializing the meta-patches via *data initialization* leads to higher universal adversarial accuracy compared to *random initialization*. The middle plot shows that the model trained with targeted meta-patches is more robust than the model trained with untargeted meta-patches, because targeted meta-patches allow for a greater diversity. The right plot shows results of randomly choosing a patch ($F = 1$), selecting the worst patch from $F = 5$ samples, and selecting the worst patch from $F = 10$ samples. Training the model with more than one sample ($F > 1$) improves the model's robustness but robustness saturates for $F = 10$ while larger $F$ increases computational cost.

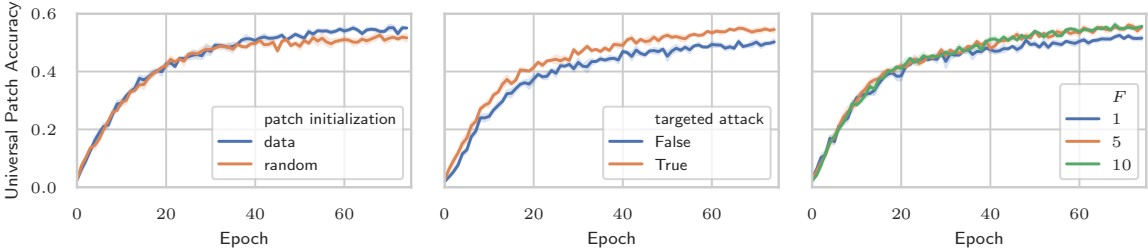

*Figure 2.* Comparison of MAT models' learning curves against the transfer attack for generating universal patches. (Left) meta-patch initialization in MAT between *data initialization* and *random initialization*. (Middle) MAT using targeted attacks and untargeted attacks for updating meta-patch. (Right) MAT uses the worst meta-patch chosen from different numbers of samples $F$.

Figure 3 shows learning curves of ablated versions of MAT against the transfer attack. In accordance with Table 1, training with a larger number of meta-patches $P$, more iterations in I-FGSM, and with a REPTILE learning rate smaller than 1.0 consistently improves robustness.

While Table 1 shows the worst accuracy of a setting against all attacks of the grid search, Figure 4 summarizes the accuracy of all attacks of the grid search in a box plot. Each value is averaged over 5 independent runs with 5 different seeds for each training procedure. Each model is evaluated against three patch attack procedures: *data*, *random*, and *low frequency*. Configurations with large variance indicate that the model might appear to be robust if hyperparameters of the attack are chosen badly. This effect is particularly pronounced for AT and SAT against the random initialized S-PGD attack, where only very few attack configurations are able to strongly degrade performance.

Moreover, the results exhibit that MAT is the only model robust against the data initialization attacks. None of the attacks reduce MAT's accuracy below 0.5 regardless of initialization methods. As discussed before, attacks through data initialization are more effective than through random initialization and attacks employing a low frequency filter are most effective on MAT with random initialization (MATr). Nevertheless, MATr still shows stronger robustness than all other approaches except MAT.

**Patch-RS**  Besides the diverse set of gradient-based attacks, we additionally conduct an experiment with the non-gradient based Sparse-RS attack with the Patch-RS sampler (Croce et al., 2020). The results are summarized in Table 7. MAT outperfoms other methods under this attack, indicating that superior performance under gradient-based attacks was not stemming from gradient-masking.

|  | Standard | AT | SAT | UAT | MAT |
|---|---|---|---|---|---|
| Patch-RS Acc. | 0.048 | 0.229 | 0.255 | 0.267 | 0.569 |

*Table 7.* Accuracy on Tiny ImageNet against a Patch-RS attacker.

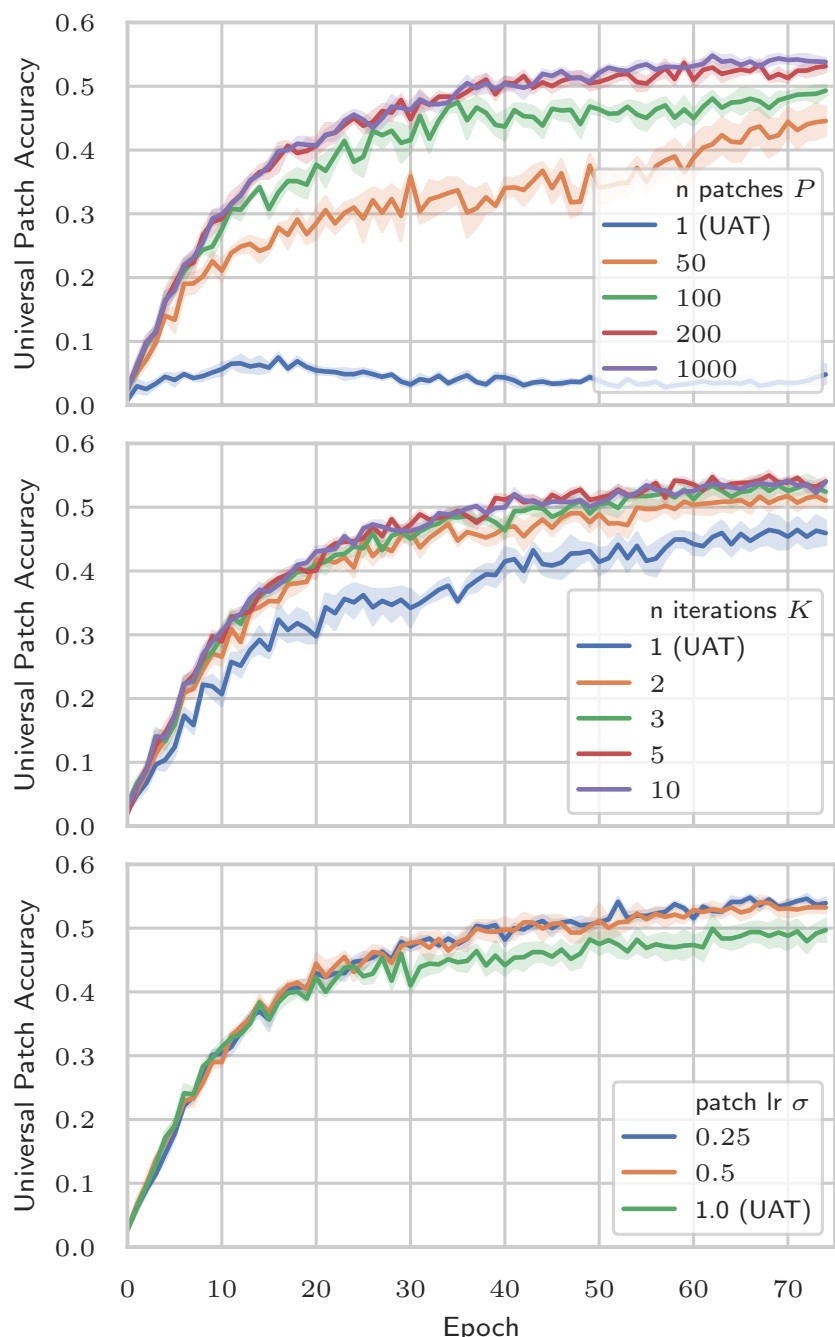

*Figure 3.* Ablation study of MAT in the aspects where it differs from UAT (Shafahi et al., 2018). Shown is accuracy on Tiny ImageNet against a patch generated with the transfer attack: UAT learns a single perturbation/patch while MAT learns an ensemble (upper plot). UAT performs one iteration of I-FGSM while MAT performs multiple (middle). UAT uses joint training ($\sigma = 1$), while MAT uses full REPTILE with $\sigma \le 1$ (bottom plot).

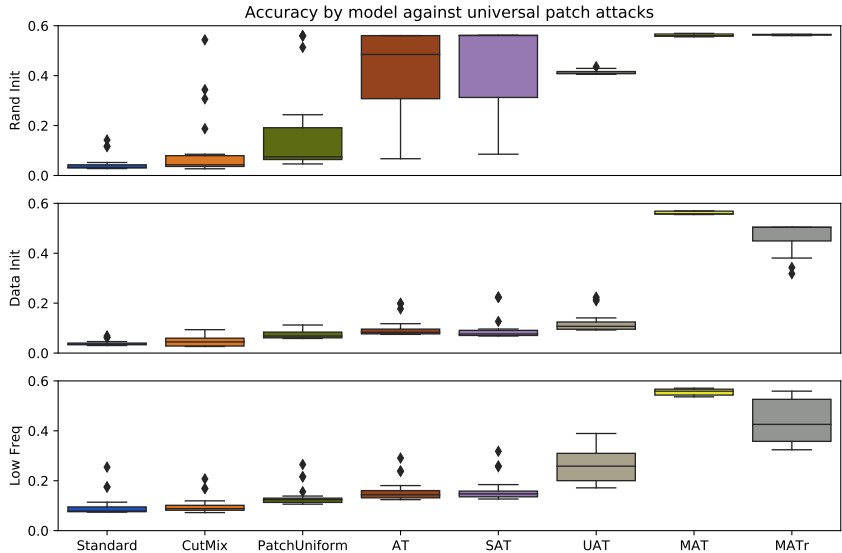

*Figure 4.* Robustness against universal patch attacks: results correspond to Table 1 but show distribution of accuracy over elements of the grid search rather than only the worst accuracy. Each value is averaged across 5 runs (with 5 different seeds) for each configuration per training approach. MATr is the MAT model trained with randomly initialized meta-patches, whereas MAT represents the model trained with meta-patches through data initialization. The rows correspond to three different universal patch attacks.

**Patch Size**    Figure 5 shows the adversarial accuracy for different patch sizes. MAT outperforms all other methods for arbitrary patch sizes smaller than 40x40.

### D.1.2. ROBUSTNESS AGAINST UNIVERSAL PERTURBATION ATTACKS

We present an analogous evaluation as in Section 4.1 for a universal perturbation attack. We use the same dataset, neural architecture, and training pipeline but train the models specifically for universal perturbation attacks. We allow universal perturbations $\xi$ with $||\xi||_\infty \leq 20/255 \approx 0.078$. In comparison with training models against universal patch attacks, the key difference is that the models are trained specifically for universal perturbation attacks instead of universal patch attacks. Following the training configuration shown in Table 8, we train ResNet V1 models with 4 training approaches - Standard, AT, UAT, and MAT. We do not present results for SAT (Mummadi et al., 2019) since we have not found a stable configuration of hyperparameters for this setting; however, the results of Mummadi et al. (2019) indicate that SAT should perform slightly better than AT when configured appropriately. We evaluate the robustness of the models against the same attacks as for patch attacks.

| SETTING | AT | UAT | MAT |
|---|---|---|---|
| PATCH INITIALIZATION | RANDOM | RANDOM | RANDOM |
| WORST OVER $F$ SAMPLES | 1 | 1 | 5 |
| REPTILE LEARNING RATE $\sigma$ | 0 | 1 | 0.25 |
| NUMBER OF META-PERTURBATIONS $P$ | — | 1 | 1000 |
| I-FGSM STEP-SIZE $\alpha$ | [0.0001, 0.02] | 0.01 | [0.0001, 0.02] |
| I-FGSM ITERATIONS $K$ | 5 | 1 | 5 |

*Table 8.* Configuration of MAT with different approaches against universal perturbation attacks for image classification.

The evaluation results are summarized in Table 9. In comparison with the results of universal patch attacks in Table 1, we notice a few interesting differences: firstly, clean accuracy is degraded for all variants of adversarial training compared to standard training. This indicates a trade-off between clean performance and robustness in this threat-model. Secondly, in contrast to standard training, AT and UAT made non-trivial gains in robustness, whereas their robustness did not improve against universal patches addressed in Section 4.1. Thirdly, the accuracy of UAT in Table 9 shows that UAT overfits

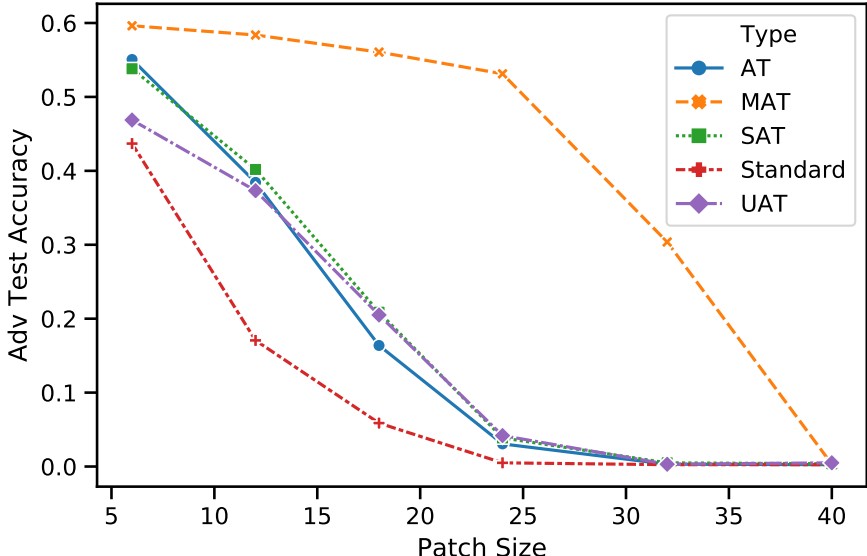

*Figure 5.* Effect of patch size on adversarial accuracy on Tiny ImageNet.

| SETTING | CL | RI | DI | LF | TR | MIN |
|---|---|---|---|---|---|---|
| STANDARD | **0.55** | 0.03 | 0.04 | 0.03 | 0.03 | 0.03 |
| AT (MADRY ET AL., 2018) | 0.48 | 0.33 | 0.26 | 0.33 | 0.27 | 0.21 |
| UAT (SHAFAHI ET AL., 2018) | 0.46 | 0.23 | 0.23 | 0.23 | 0.17 | 0.17 |
| MAT (FULL) | 0.48 | **0.42** | **0.42** | **0.42** | **0.39** | **0.39** |

*Table 9.* Accuracy (mean over 5 runs) of different methods against universal perturbation attacks on Tiny ImageNet.

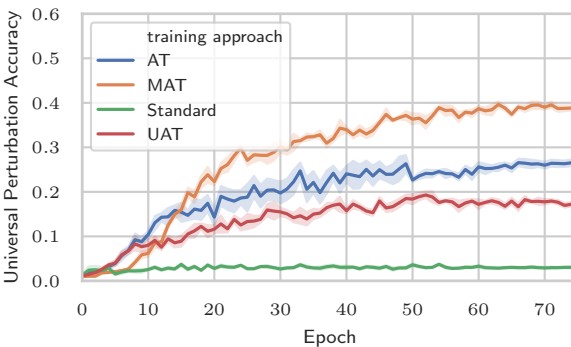

*Figure 6.* Comparison of learning curves of Standard training, AT, UAT, and MAT against the transfer attack for generating universal perturbations.

less strongly to the randomly initialized S-PGD attack compared to the universal patch attacks in Table 1. Despite these differences, MAT considerably outperforms all other methods in terms of robustness also in this setting.

Figure 6 shows the learning curves of those training approaches against the transfer attack for generating universal perturbations. Notably, while MAT is less robust in the early phase of training, it reaches a significantly higher level of robustness in the end.

Figure 7 shows the box plot corresponding to Table 9. The accuracy of MAT models is above 0.4 for all three attacks and shows little variance. In contrast, UAT and AT are robust against certain attack configurations but against an optimally configured attack, accuracy degrades to 0.25 or less. This shows that evaluating robustness reliably requires a strong set of attacks and their well-tuned hyperparameters.

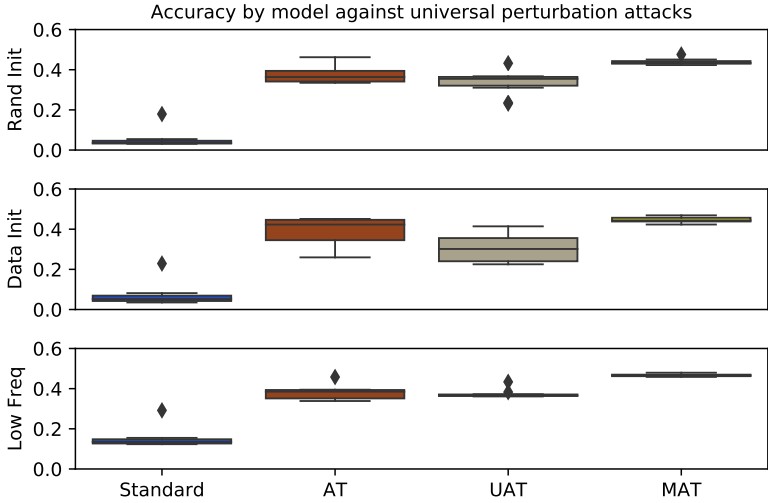

*Figure 7.* Robustness against universal perturbation attacks: results correspond to Table 9 but show distribution of accuracy over elements of the grid search rather than only the worst accuracy. Each value is an average across 5 runs (with 5 different seeds) for each configuration per training approach. The rows correspond to three different universal patch attacks.

### D.2. Object Detection on Bosch Small Traffic Lights Dataset

We present more detailed results for the experiment reported in Table 2: Table 10 shows results for the different attacks conducted on the respective models. Figure 8 shows corresponding boxplots for recall and mAP, respectively. Notably, only the recall of the standard model can be reduced considerably (meaning true positives can be hidden) and this requires an

*Table 10.* More detailed results corresponding to Table 2: Mean recall (left) and mean average precision (right) of different methods on clean data (CL) and against universal patch attacks with random init (RI), init with a cropped image patch (DI) and low-frequency filter (LF) on Bosch Small Traffic Lights (Behrendt & Novak, 2017).

| Recall | CL | RI | DI | LF | MIN | mAP | CL | RI | DI | LF | MIN |
|---|---|---|---|---|---|---|---|---|---|---|---|
| STANDARD | **0.47** | 0.37 | 0.38 | 0.43 | 0.37 | | **0.41** | 0.09 | 0.10 | 0.16 | 0.09 |
| UAT | **0.47** | **0.45** | **0.45** | **0.45** | **0.45** | | **0.41** | **0.40** | 0.29 | 0.26 | 0.26 |
| MAT (DEFAULT) | 0.45 | 0.43 | 0.43 | 0.43 | 0.43 | | **0.41** | 0.39 | 0.35 | 0.35 | 0.35 |
| (+DATA INIT) | 0.45 | 0.43 | 0.43 | 0.42 | 0.42 | | **0.41** | 0.38 | **0.39** | **0.39** | **0.38** |
| ($K = 1$) | 0.46 | 0.44 | 0.43 | 0.44 | 0.43 | | **0.41** | **0.40** | 0.31 | 0.27 | 0.27 |
| ($K = 1, P = 1$) | 0.46 | 0.44 | 0.43 | 0.43 | 0.43 | | **0.41** | **0.40** | 0.21 | 0.17 | 0.17 |

appropriately configured attack. Interestingly, a low-frequency attack is not effective for reducing the recall of any model. In contrast, low-frequency attacks are the most effective ones for reducing mAP, that is: for causing false positive detections. While randomly initialized S-PGD is not successful at reducing the mAP of any model besides the standard model, many low-frequency attacks of varied attack configurations reduce mAP of most models (except MAT + data) considerably. In contrast, S-PGD from data initialization can be effective but fails in most cases to reduce mAP for all but the standard model.

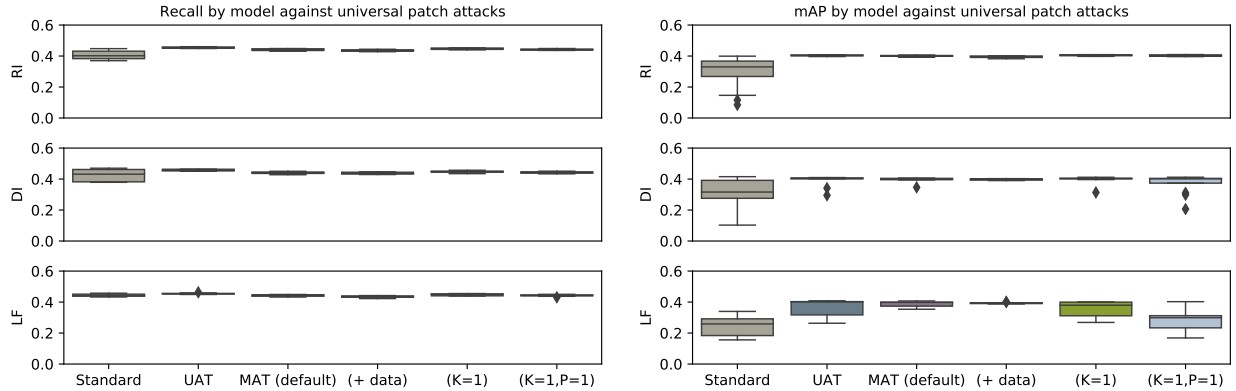

*Figure 8.* Recall (left) and mean Average Precision (right) by model against universal patch attacks on Bosch Small Traffic Lights Dataset. MAT (default): meta-patch is initialized uniform-randomly. MAT (+data): meta-patch is initialized from a cropped image. Both MAT (default) and MAT (+data) have I-FGSM iteration $K=3$ and number of patches $P = 10$. MAT($K=1$): I-FGSM iteration $K=1$ and number of patches $P = 10$. MAT($K = 1, P = 1$): I-FGSM iteration $K=1$ and number of patches $P = 1$.

## E. Illustration of Patch Attacks on Tiny ImageNet

We illustrate universal patch attacks on models trained on Tiny ImageNet in Figure 9. Note that these are the strongest patches found against these models during the grid search. Oftentimes, the generated patch resembles the target class: examples for this are the low-frequency attack on a standard model (fooling it to mistake a chimpanzee for a police van), the random initialization attack against the SAT model (fooling it to mistake the chimpanzee for a ladybug), the data initialization attack against the SAT model (fooling it to mistake the chimpanzee for an orange), or the low-frequency attack against the UAT model (fooling it into classifying the input as a fire salamander based on the characteristic texture of the patch). While these misclassifications can be explained, a human would very likely still classify the inputs as chimpanzees. Attacks on MAT (full) fail to generate interpretable patches; however, transferring patches generated for other models (such as the shown ones) to MAT does not cause misclassifications either.

## F. Illustration of Patch Attacks on Bosch Small Traffic Lights Dataset

We illustrate universal patch attacks on models trained on Bosch Small Traffic Lights Dataset in Figure 10. Note that these are the strongest patches found against these models during the grid search in terms of the mAP. These patches often invoke

**Rand Init**  **Data Init**  **Low Freq**

**Standard**

pop bottle  lifeboat  police van

**CutMix**

pop bottle  bucket  police van

**PatchUniform**

pop bottle  acorn  beer bottle

**AT**

beer bottle  chimpanzee  chimpanzee

**SAT**

ladybug  orange  chimpanzee

**UAT**

chimpanzee  acorn  European fire salamander

**MAT (full)**

chimpanzee  chimpanzee  chimpanzee

high confidence false detections. However, MAT with data initialization does not show any false positives.

For the patches found for MAT (Data Init), we show the progress of the patches during the attack in Figure 11. Similarly, Figure 12 shows the patches' evolution during an attack on the standard model. Note that patches converge fairly quickly, namely, running attacks longer would not make them stronger. Moreover, all three patches for MAT converge to a red-cyan pattern and the patches for data and random initialization exhibit very similar patterns. This indicates that this pattern is actually a minimizer of the loss with a large basin of attraction. However, as Figure 10 shows, it does not really fool the model. Finally, Figure 13 shows the training of the patch shown in Figure 1.

## G. Illustration of Perturbation Attacks on Tiny ImageNet

We illustrate universal perturbation attacks on models trained on Tiny ImageNet in Figure 14. Note that these are the strongest perturbations found against these models during the grid search.

## H. Related Work

We review work on generating universal perturbations/patches, defending against them, and meta-learning.

**Generating Universal Perturbations**   Adversarial perturbations are changes to the input that are crafted with the intention of fooling a model's prediction on the input. Universal perturbations are a special case in which one perturbation needs to be effective on the majority of samples from the input distribution. Most work focuses on small additive perturbations that are bounded by some $\ell_p$-norm constraint. For example, Moosavi-Dezfooli et al. (2017) proposed the first approach by extending the DeepFool algorithm (Moosavi-Dezfooli et al., 2016). Similarly, Metzen et al. (2017) extended the iterative fast gradient sign method (Kurakin et al., 2017) for generating universal perturbations on semantic image segmentation. Mopuri et al. (2017; 2018) presented data-independent attacks and Hayes & Danezis (2018) proposed using a generative model for learning a diverse distribution of universal perturbations. Li et al. (2019) presented a physical-world attack in which a translucent sticker is placed on the lens of a camera, which adds a universal perturbation to the image taken by the camera, and showed that this can fool an image classification system.

Other types of universal perturbations are so-called adversarial patches (Brown et al., 2017). In these universal patch attacks, the adversary can arbitrarily modify a small part of the image, typically a connected rectangular area, while leaving the remaining part of the image unchanged. Following Athalye et al. (2018), randomizing conditions such as location, rotation, scale, and lighting during the attack can make the universal patch sufficiently effective to fool the model when it is printed out and placed in the physical world. Later work has generalized these physical-world attacks to object detection (Lee & Kolter, 2019; Huang et al., 2019) and optical flow estimation (Ranjan et al., 2019).

**Defending Universal Perturbations**   First works for defending against universal perturbations are based on training a model against a fixed or slowly updated set/distribution of universal perturbations: Moosavi-Dezfooli et al. (2017) precompute a set of universal perturbations that are used during training, Hayes & Danezis (2018) learn a generative model of universal perturbations, and Perolat et al. (2018) build a slowly increasing set of universal perturbations concurrent to model training. A shortcoming of these approaches is that the model might overfit the fixed or slowly changing distribution of universal perturbations. However, re-computing universal perturbations in every mini-batch from scratch is prohibitively expensive. To address this issue, SAT Mummadi et al. (2019) trains a model against so-called shared perturbations. These shared perturbations do not have to be universal but only need to fool the model on a fixed subset of the batch. However, since the shared perturbations are recomputed in every mini-batch, it assumes a few gradient steps are sufficient to find strong perturbations from random initialization. In contrast, our method meta-learns strong initial perturbations. In UAT (Shafahi et al., 2018), training the neural network's weights and updating a single universal perturbation happen concurrently, which scales to a large dataset. However, our experiments in Section 4 indicate that a single incrementally and slowly updated perturbation is not sufficiently strong and diverse for making a model robust against all possible universal perturbations. Instead, our method meta-learns a large and diverse collection of perturbations during training.

For defending against adversarial patches, Chiang et al. (2020) proposed an approach of extending interval-bound propagation (Gowal et al., 2019) to the patch threat model. While this allows certification of robustness, it only scales to tiny patches and reduces clean accuracy considerably. Wu et al. (2020) proposed the "defense against occlusion attack", which applies adversarial training to inputs perturbed with input-dependent adversarial patches placed at specific positions determined, for

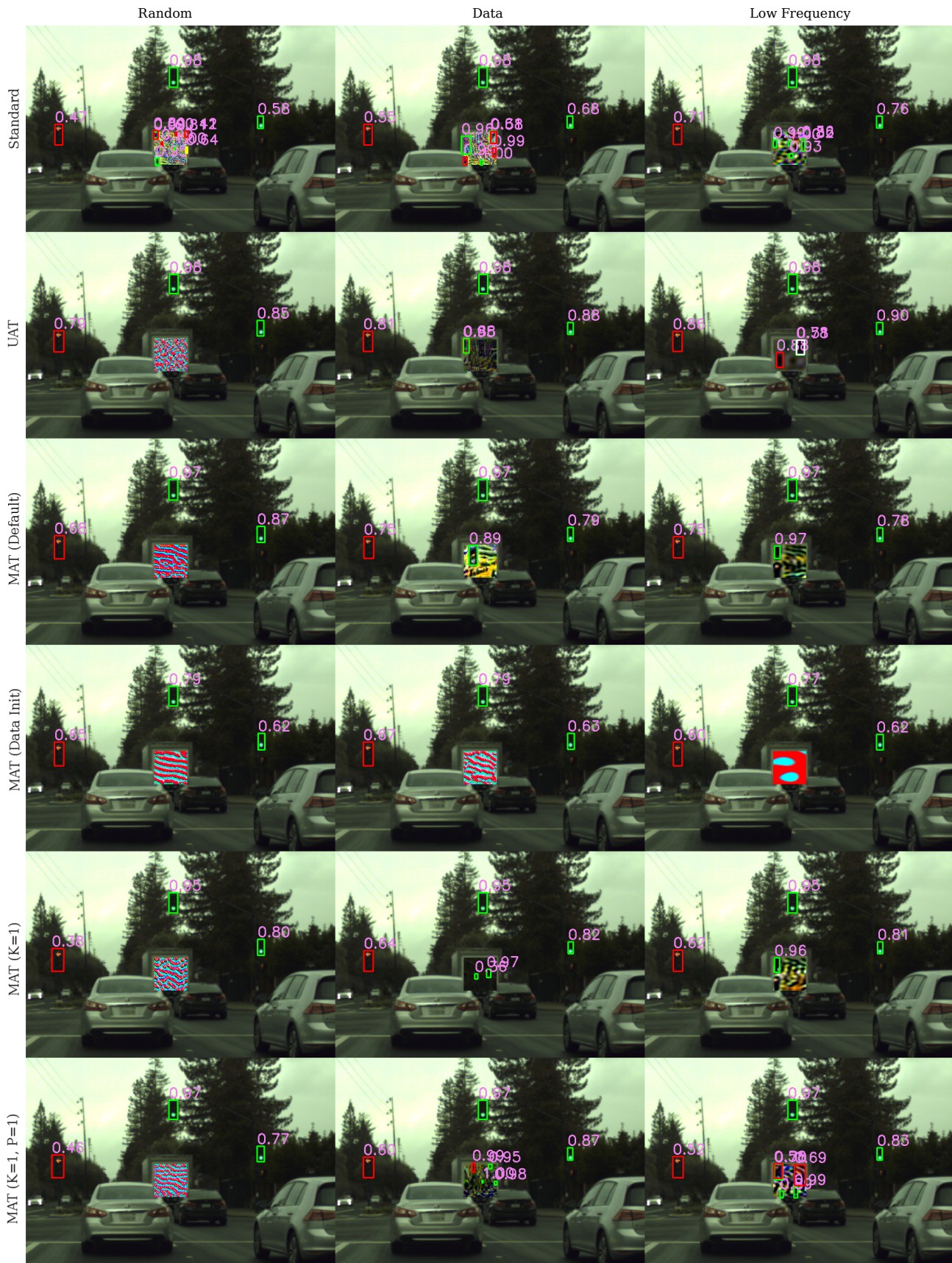

*Figure 10.* Best (in terms of mAP) random initialization, data-crop initialization and low-frequency attacks against the models from Table 2. The patch location is fixed for a better comparison.

Data, Learning Rate: 0.01

Random, Learning Rate: 0.1

Low-Frequency Cutoff: 100, Learning Rate: 0.01

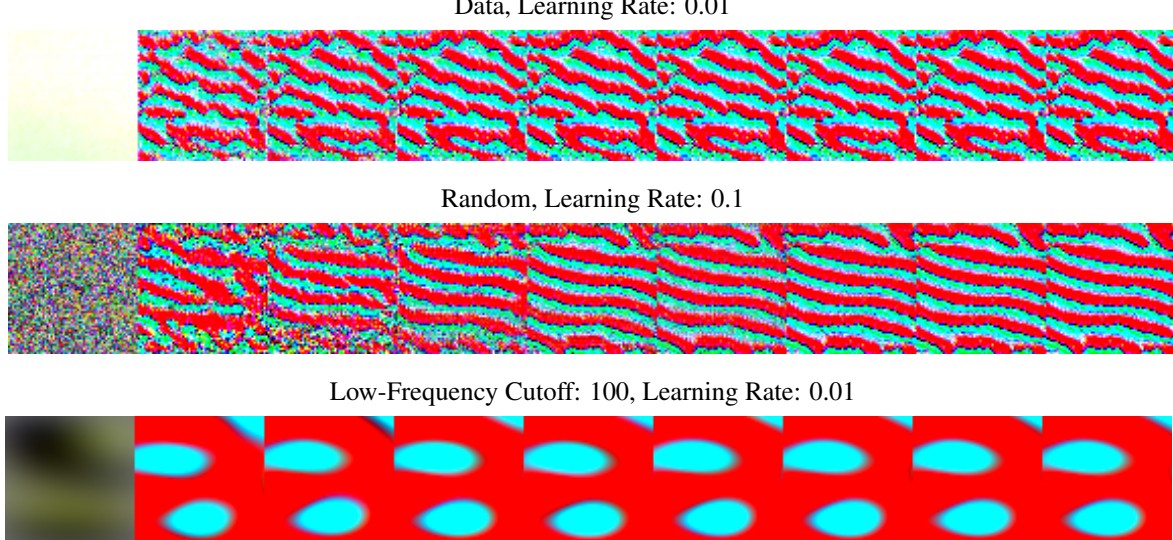

*Figure 11.* Training of patches against a MAT (Data Init) model. Attack from data-crop initialization with step size of 0.01 (top). Attack from random initialization with step size of 0.1 (center). Low-frequency attack with cutoff frequency 100 from data-crop initialization with step size of 0.001.

Data, Learning Rate: 0.01

Random, Learning Rate: 0.01

Low-Frequency Cutoff: 100, Learning Rate: 0.001

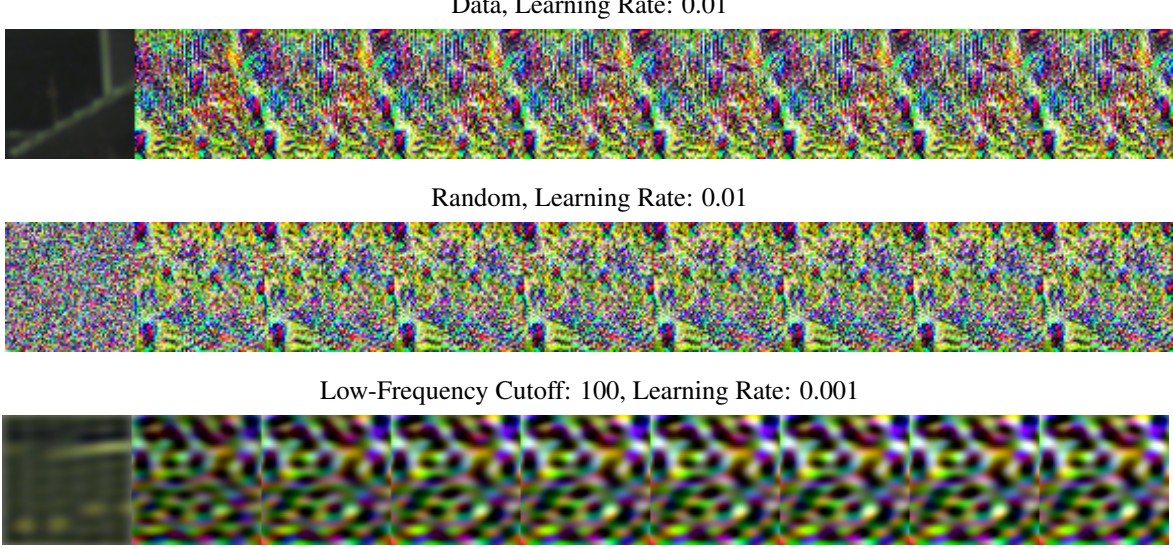

*Figure 12.* Training of patches against standard model: Attack from data-crop initialization with step size of 0.01 (top). Attack from random initialization with step size of 0.01 (center). Low-frequency attack with cutoff frequency 100 from data-crop initialization with step size of 0.001

Low-Frequency Cutoff: 100, Learning Rate: 0.001

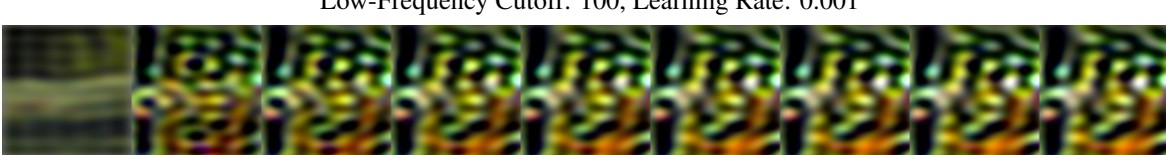

*Figure 13.* Training of patch from Figure 1. This patch is generated with a low-frequency attack with cut-off frequency 100, learning rate of 0.001 and starting from data-crop initialization.

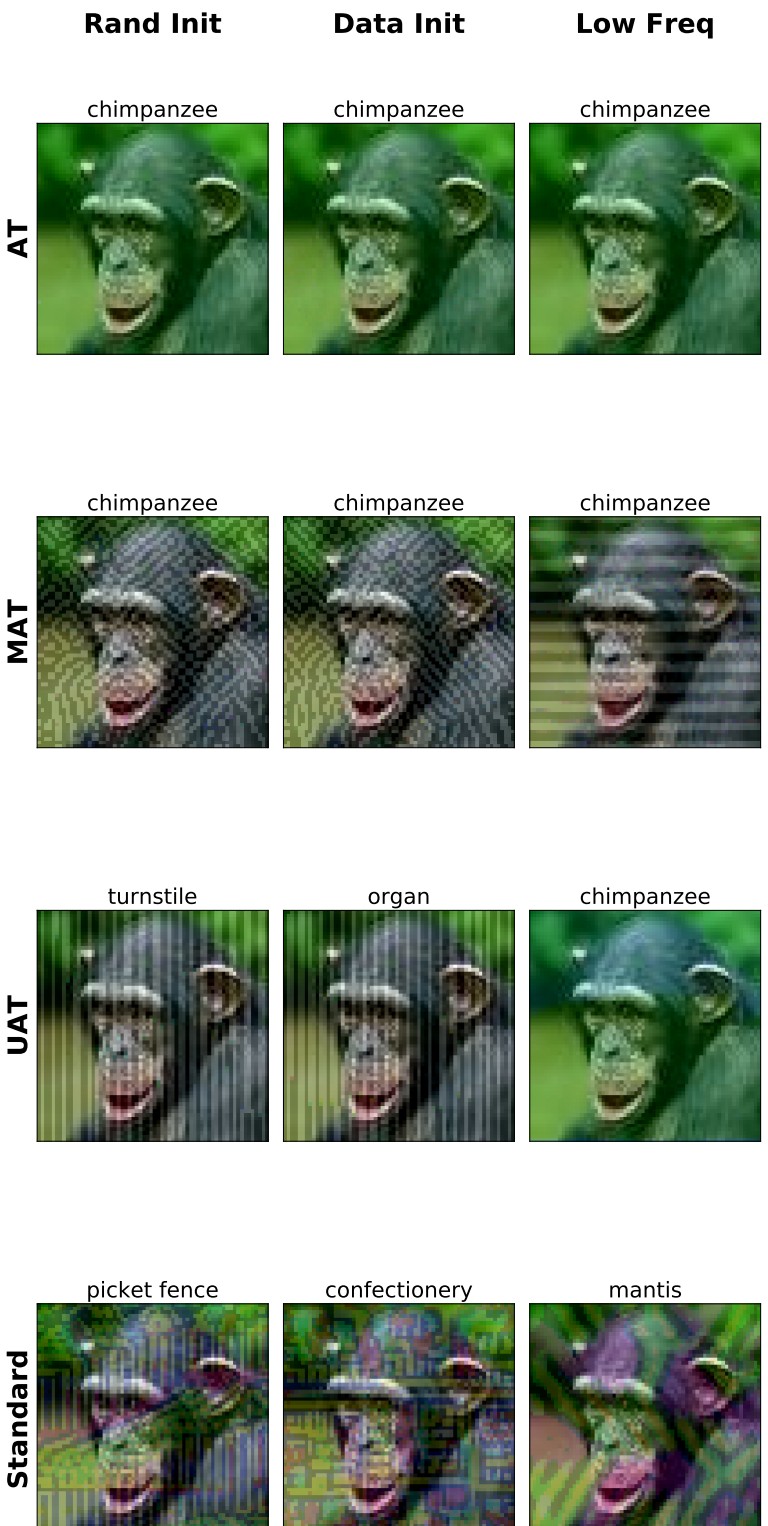

*Figure 14.* Best random initialization, data initialization, and low-frequency perturbation attacks against the respective models from Table 9. The model prediction is given above each plot. The correct label is *chimpanzee*.

example, by the input gradient magnitude. Since they generate patches from scratch, they require an expensive optimization of the patch for every training batch. Moreover, robustness against stronger attacks such as those proposed in Section B remains unclear. Saha et al. (2019) hypothesize that vulnerability of object detectors against adversarial patches stems from contextual reasoning. Accordingly, they propose Grad-defense which penalizes strong dependence of object detections on their context in a data-driven manner, where dependence is determined by Grad-CAM (Selvaraju et al., 2019). Lastly, some non-adversarial data augmentation techniques resemble the universal adversarial patch scenario: they add a Gaussian noise patch (Lopes et al., 2019) or a patch from a different image (CutMix) (Yun et al., 2019) to each input. CutMix is conceptually very similar to the out-of-context defense (Saha et al., 2019). However, as demonstrated in our experiments in Section 4, even though these approaches increase robustness against occlusions, they are unlikely to increase robustness against universal patch attacks.

**Meta-Learning**   Gradient-based meta-learning methods such as MAML (Finn et al., 2017) or REPTILE (Nichol et al., 2018) allow learning initial parameters for a class of optimization tasks, so that one can find close-to-optimal parameters on a novel task from the distribution with a small number of gradient steps. Moreover, meta-learning can also be used to learn the task optimizer itself such as by Xiong & Hsieh (2020) in the context of adversarial training. While it is common to meta-learn initial weights for neural networks, we propose that these algorithms can also be used to meta-learn initial values for universal perturbations. In this work, we combine REPTILE with adversarial training because of the low computational overhead of REPTILE; however, in principle other gradient-based meta-learning methods could also be used as part of our method.