# OpenReview forum: "Meta Adversarial Training against Universal Patches"
_ICML.cc/2021/Workshop/AML — ICML 2021 Workshop AML Poster_

### Official Review · Reviewer_upAi · 2021-06-20
**Interesting work**

**Rating:** Accept
**Confidence:** 5

**Review:**

This paper proposes the meta adversarial training (MAT) method, which meta learns an initialization and uses I-FGSM to craft adversarial patches during training. This strategy is more efficient compared to previous methods and is empirically verified on different datasets. MAT can effectively defend against physical attacks and is worthy of more attention from the community.

---

### Decision · Program_Chairs · 2021-06-21

**Decision:**

Accept (Poster)

**Comment:**

This paper proposes MAT for training robust models against patch attacks. The effectiveness is verified on different datasets.